# Validity Study of the Spanish Version of the Parental Stress Index (PSI-4) in Its Two Forms (Long Form and Short Form)

**DOI:** 10.3390/children12111466

**Published:** 2025-10-29

**Authors:** Mercedes Ríos-Requeni, Yurena Alonso-Esteban, Esperanza Navarro-Pardo, Francisco Alcantud-Marín

**Affiliations:** 1Centros de Estudios Marni (Valencia), 46019 Valencia, Spain; mercedesrios@marni.es; 2Department of Developmental and Educational Psychology, University of La Laguna, 38200 San Cristóbal de La Laguna, Spain; 3Department of Developmental and Educational Psychology, University of Valencia, 46010 Valencia, Spain; esperanza.navarro@uv.es (E.N.-P.); francisco.alcantud@uv.es (F.A.-M.)

**Keywords:** construct validity, instrument adaptation, parental stress, parental stress index 4th edition (PSI-4), psychometric date

## Abstract

**Highlights:**

•The PSI-4 is the standard instrument for measuring parental stress.•The Spanish version of the PSI-4 (PSI-4/SP) shows similar evidence of validity to the English version.

**What are the main findings?**
•The strengths of the PSI-4/SP are its internal consistency and concurrent and discriminant validity.•The fit of the PSI-4 model is improved by subdividing the Parent Domain into two factors (personal and situational).

**What are the implications of the main findings?**
•The PSI-4/SP is suitable for detecting dysfunctional families regardless of the reason for the dysfunction.•The use of the long version of the PSI-4/SP enables the study of potential reasons for parental dysfunction and facilitates the development of more effective interventions.•The official adaptation of the PSI-4 to the Spanish population will enable the development of cross-cultural studies of the parental stress construct.

**Abstract:**

**Background/Objectives:** Parental stress is caused by the accumulation of difficulties related to raising children and is directly related to the educational functioning of a family and family dysfunction. The PSI (Parental Stress Index) is an assessment instrument and is considered the gold standard for measuring parental stress; however, there is no official Spanish version of this tool. **Methods**: Both the PSI-4 long form and short form were administered to a sample of 828 parents of children between 0 and 12 years of age. **Results:** The discrepancy between the scores of the original typification sample and the Spanish sample necessitates the establishment of specific normative values. Regarding internal consistency, test–retest stability, and construct validity, the results demonstrate consistency with the original study and other adaptation studies. The validity of the theoretical structure of the instrument was studied using confirmatory factor analysis (CFA). The CFA fit indices in the overall PSI-4 LF model (101 items distributed across 13 first-order factors and two second-order factors) are not satisfactory. Therefore, the methodology used by the author in constructing the PSI-4 was employed, which involved performing two exploratory factor analyses (EFA). The domains of both parents and children exhibit partial replication of the theoretical subscales of the PSI-4. The analysis of the short form of the PSI-4 SF (comprising 36 items distributed across three subscales and a total score) yielded results similar to those of the long form. **Conclusions:** These results suggest the potential for developing specific standards for the Spanish population and conducting criterion validity studies for the clinical use of all dimensions of the PSI-4 in its two forms (LF and SF).

## 1. Introduction

Parental stress refers to the psychological reaction that parents may experience when attempting to fulfil their caregiving and educational responsibilities. It is generally perceived as a negative or aversive response to the demands and expectations of parenting, especially when these are not met [1,2]. Parental stress relating to child-raising is very common [3,4]. It is estimated that between 36% and 50% of parents experience some level of concern or stress relating to their children’s behaviour, health, and development [4,5]. Sometimes, this requires professional counselling and support [5]. There is consistent evidence of an association between parental stress and various issues affecting parents, children, and families. These issues include parental mental health and psychological well-being; emotional and behavioural problems in children; academic under-achievement; and inappropriate parenting behaviours, such as child abuse, marital problems, and family dysfunction [6].

According to the general stress model proposed by Lazarus and Folkman [7], stress results from an interaction between stressors, cognitive assessment of competency, and coping responses. Based on classic works by Selye [8,9,10,11], among others, we can also assume that stress is the sum of all factors. Parental stress differs conceptually from other types of stress experienced by parents (known as life stress), such as stress related to work, financial problems, and negative life events. However, the presence of other life stressors can exacerbate parental stress.

In their systematic review, Fang et al. [12] found no correlation between parental stress and either the mother’s age or the child’s gender. However, they observed relationships with maternal depression, general child-related problems, social support, and parents’ educational level. The level of parental stress fluctuates depending on the stage of child development [13,14]. Conversely, it has been hypothesised that stressors are multidimensional regarding both their source and their type [15]. This characteristic makes it challenging to develop explanatory theoretical models that facilitate forecasting. Traditionally, models are built on the principle of factor orthogonality. However, in the case of stress in general, and parental stress in particular, factors interact with each other (because they are interdependent) [16,17]. Abidin [18] developed a model of parent–child relationships to explain parental stress. This model is composed mainly of two broad dimensions: the characteristics (or domain) of children and the characteristics (or domain) of parents.

Fang et al. [12] introduced a new interpretation of Abidin’s model, in which factors associated with total parental stress are organised into three domains: parents, children, and situation. The Parent Domain encompasses personality and functional aspects, including depression, attachment, and feelings of competence. The Child Domain refers to temperament and behaviour, while the situational domain includes social support, isolation, role restriction, and marital relationships. Figure 1 shows the model proposed by Fang et al. [12], which is a reorganisation of the variables proposed by Abidin. As can be seen in the figure, total parental stress is made up of three components: Child Factors, the parent’s Personal Factors, and Situational Factors. Please note that this reorganisation divides Abidin’s original parental dimension into two factors: personal and situational. Additionally, the inter-relationship of these three dimensions is indicated by discontinuous lines, showing that they can modulate the importance of their components beyond the accumulation of factors. For example, a parent’s feelings of competence (CO) are generated and reinforced by positive feedback from the child’s development (RE). Similarly, parental attachment (AT) can be generated and reinforced by any of the child’s characteristics, or even by the child’s own attachment to their parents; these inter-relationships make it challenging to develop an explanatory model of parental stress. This model served as the basis for the development of the Parental Stress Index (PSI) by Abidin [19].

Although several scales are used to assess parental stress, the PSI is the oldest and most frequently used [6,20,21]. It has been used since the first version was developed by Abidin [19] and is currently in its fourth edition [22].

Since the first version of the PSI was published, several studies have shown good internal consistency and adequate test–retest reliability. As it has evolved, different versions have been translated and administered in various countries and validated in different cultural groups, including France [23], Portugal [24], Canada [25], Finland [26], the Netherlands [27], and China [28,29].

The original PSI consists of 101 items, divided into 13 factors organised into two domains: parents and children. In response to demand from researchers and clinicians, and following various factorial studies [30], an abbreviated version (PSI-SF) was published. This focuses on the key element of the parent–child system, simplifying the structure into three domains: Parental Distress (PD), Parent–Child Dysfunctional Interaction (P-CDI) and Difficult Child (DC). The number of items in each domain is balanced, totalling 36 items.

A substantial body of literature has been compiled, providing ongoing evidence of its clinical usefulness and sound psychometric properties [6,31,32,33,34,35,36,37,38]. However, most validation studies have only focused on the short version, which limits the development of research [20].

Several changes have been made to the wording of the items in the current version (PSI-4) to reflect new family and parenting styles, including single-parent families, same-sex families, and heterosexual families. There has also been a significant change to the instructions section. For example, it explains that the assessment is made in relation to a specific child, rather than to parenting situations in general; this allows us to capture information about the parental stress generated by raising each child, rather than providing a general index of parental stress. There are two versions of the PSI-4: PSI-4 LF—an evolution of the original Parental Stress Index [39]—and PSI-4 SF. The PSI-4 LF includes 13 subscales (6 in the child subscales and 7 in the parent subscales) in two domains (Parent and Child domains). The short version (PSI-4 SF), derived from the previous one [30], consists of 36 items organised around three dimensions (Parent Domain, Child Domain, and Dysfunctional Parent–Child Interaction). The PSI-4 SF is a valid instrument for detecting parenting dysfunctions, while the PSI-4 LF can help to formulate hypotheses about parenting and, thereby, design an intervention programmed. In addition, the PSI-4 can monitor the effects of the intervention. The aim of this study is to analyse the psychometric characteristics of the Spanish version of the PSI-4 in its two versions.

## 2. Materials and Methods

### 2.1. Procedure

In Spain, most public nursery and primary schools (for children aged 3 to 12 years old) have parents’ associations, such as AMPAS (Association of Parents of Infant and Primary Schools), that play an active role in organising and promoting school activities. Early childhood education centres for children aged 0 to 2 are typically private and operate under a different dynamic.

To reach a sufficiently large and representative sample of parents, a dual recruitment method was chosen. We contacted the AMPAS and the educational guidance services at schools. Once participation had been agreed upon, we sent them blank questionnaires and informed consent form to distribute among the members of each school association. The board of directors, responsible for participation, supervised the collection of the sealed envelopes containing the completed questionnaires and signed informed consent form and their dispatch to the research team. The volunteer parents completed the PSI-4 questionnaire, placed it in the envelope, and sealed it to ensure anonymity. As the response rate using this method was limited (less than 7% of the questionnaires distributed), the educational guidance services of schools were also contacted as a complementary measure. In these cases, once the school management had agreed to participate, blank questionnaires and informed consent form were sent to the schools, which then distributed them to parents. The schools’ guidance counsellors were responsible for collecting the sealed envelopes containing the completed questionnaires and sending them to the research team.

The Code of Ethics of the World Medical Association (the Declaration of Helsinki) was adhered to throughout the process. The anonymity of the participating parents was respected at all times. The completed questionnaires and signed informed consent forms were always submitted in sealed envelopes. In this way, we consider that we have respected the confidentiality of the information received. The research team anonymised the computer records and proceeded to destroy the paper documents. The project was approved by the Human Research Ethics Committee of the University of Valencia (No. H1536233421283). Data collection began in September 2019 but was interrupted by the confinement and quarantine measures imposed by the COVID-19 pandemic in March 2020; consequently, data collection was not resumed until January 2021 and terminated in April 2021.

### 2.2. Participants

More than one thousand questionnaires were distributed. Of these, 828 were valid (approximately 11% of the questionnaires received were excluded due to errors in the response form, incomplete demographic data, or lack of informed consent, among other reasons).

Table 1 shows the basic characteristics of the sample. The questionnaires were mostly completed by mothers (82.1%); the participation of fathers varied depending on the age of the children, ranging from a minimum of 8.8% to a maximum of approximately 30%. Regarding parental education, more than 50% of parents had a high school diploma or higher. These data suggest that the sample consists of parents with a medium level of education.

### 2.3. Instruments

Regarding translation and validation in the Spanish population, Solis and Abidin [40] and Díaz-Herrero et al. [41] translated and validated the previous version of the PSI-SF, and Hernández-Trejo et al. [42] developed the Mexican version of PSI-4 SF. Barroso et al. [31] conducted a study of a Spanish-speaking sample in the USA using PSI-4 SF. As these studies were conducted on clinical samples, a study with a larger and non-clinical sample was needed. Furthermore, as the PSI scale is a copyrighted instrument, there is currently no official Spanish translation and adaptation. For this reason, we previously contacted the author and copyright owner to translate and adapt the PSI-4 instrument in its two forms to the Spanish population.

Initially, authorisation was sought from the copyright owners (PAR and Hogrefe TEA Editions). Once it was obtained, the questionnaire was given to three bilingual psychologists with experience in counselling, and the ITC (International Testing Commission) recommendations [43,44] were followed. Once the three translators had agreed on the differences, a single translation was created and reviewed by a linguistic expert. The final version was then back-translated into English and sent to the copyright owners and authors for approval. The result was a 101-item questionnaire in the same order and with the same response scale as the original English version. As this questionnaire corresponds to the long version of the PSI-4 and contains the 36 items from the short form, it was not necessary to apply the short form. Once the questionnaires had been collected, they were corrected in accordance with the rules outlined in the Manual [22].

#### 2.3.1. Parental Stress Index (PSI)

Abidin [19] developed the PSI as a tool comprising two independent domains or scales, each of which was a separate questionnaire, together forming a total Parental Stress Index. With regard to the characteristics of the child (Child Domain, CD), there are six subscales:Distractibility/hyperactivity (DI): This subscale is associated with stressful situations arising from a child’s inability to pay attention or from hyperactivity. Examples include often leaving tasks unfinished, difficulty concentrating on a single task, hyperactivity, restlessness, distractibility, and lack of concentration and attention. There is a total of nine items on this subscale, each of which is scored on a five-point Likert scale ranging from ‘strongly agree’ to ‘strongly disagree’.Adaptability (AD): This is associated with the stress that parents may experience when their children find it hard to adapt to changes in their physical or social environment. There are 11 items in total; a total of 9 of these use a five-point Likert scale ranging from ‘strongly agree’ to ‘not sure’ to ‘strongly disagree’, 1 uses a four-point scale ranging from ‘very easy’ to ‘very difficult’ to describe how easy or difficult it is to calm the child down after they have become upset, and the other uses a five-point scale to describe the different levels of difficulty in doing or stopping something.Reinforces parent (RE): This subscale relates to a parent who does not perceive their child’s development as something positive, or who feels rejected by them, which can jeopardise the bond between them. There are six items in total: five use a five-point Likert scale (strongly agree/not sure/strongly disagree), and one uses a four-point scale to describe how often their child plays with their parents.Demandingness (DE): This is associated with the feeling of being overwhelmed experienced by parents due to excessive demands from their child, such as excessive crying and dependency. There are a total of nine Likert-type items, each scored on a five-point scale ranging from ‘strongly agree’ to ‘strongly disagree’.Mood (MO): This subscale assesses parents’ perception of their child’s mood when the child exhibits dysfunctional emotional functioning. There are five items in total: four use a five-point Likert scale ranging from ‘strongly agree’ to ‘strongly disagree’, and one uses a five-point scale to express the child’s level of crying and agitation according to parental expectations.Acceptability (AC): Occasionally, a child’s physical, intellectual, or emotional characteristics do not match their parents’ expectations. In this case, the parents may or may not accept the child as they are, which may affect their relationship. This subscale contains seven items, each of which is scored on a five-point Likert scale ranging from ‘strongly agree’ to ‘strongly disagree’.

Regarding parental characteristics (Parent Domain, PD), the PSI develops seven sections or subdomains:Competence (CO): This subscale analyses parental self-assessment of competence. It correlates strongly with levels of parental education. This subscale contains 13 items, 7 of which are scored on a five-point Likert scale ranging from ‘strongly agree’ to ‘strongly disagree’. The remaining five items are scored on a five-point scale and cover the child’s understanding, self-assessment as a parent, and level of competence. Two items cover the educational level of the father and mother.Isolation (IS): This subscale is based on the parents’ perceived lack of family or social support. This isolation may also be related to dysfunctional, cold, distant and/or unsupportive marital relationships. This subscale contains six items, each of which is scored on a five-point Likert scale ranging from ‘strongly agree’ to ‘strongly disagree’.Attachment (AT): This subscale is associated with dysfunctional perceptions of emotional attachment to the child. It can result from problems in the child’s social and emotional development, or from the parents’ inability to understand their child’s needs and feelings. This subscale contains seven items, each of which is scored on a five-point Likert scale ranging from ‘strongly agree’ to ‘strongly disagree’.Health (HE): This subscale measures perceptions of stress-related deterioration in parents’ health. This subscale contains five items, each of which is scored on a five-point Likert scale ranging from ‘strongly agree’ to ‘strongly disagree’.Role restriction (RO): This subscale is related to parents’ perceptions of restrictions on their freedom, or frustrations they experience as parents. This subscale contains seven items, each of which is scored using a five-point Likert scale ranging from ‘strongly agree’ to ‘strongly disagree’.Depression (DP): This subscale relates to depressive states in parents that are associated with dysfunctional parenting. Some of the items on this subscale relate to feelings of guilt and unhappiness. There are nine items in total, each of which is scored on a five-point Likert scale ranging from ‘strongly agree’ to ‘strongly disagree’.Spouse/parenting partner relationship (SP): Initially, this subdomain was associated with tensions arising from differences in educational or parenting criteria within the traditional family unit. There are seven items in total, each of which is scored on a five-point Likert scale ranging from ‘strongly agree’ to ‘strongly disagree’.

The complexity of the PSI model and the need for a shorter version resulted in several factorial studies [30]. The resulting abbreviated version, the PSI-SF, focuses on the key elements of the parent–child system, simplifying the structure into three domains or subscales: Parental Distress (PD); Parent–Child Dysfunctional Interaction (PCDI); and Difficult Child (DC):Parental Distress (PD): This subscale measures the level of distress a parent experiences when raising a particular child. This distress may manifest as feelings of incompetence as a parent, restrictions imposed by parenting, conflicts with the other parent, lack of social support, or depression. The questionnaire contains 12 items, each of which is scored on a five-point Likert scale ranging from ‘strongly agree’ to ‘strongly disagree’.Parent–Child Dysfunctional Interaction (P-CDI): This subscale assesses parents’ failure to meet expectations in their role as caregivers, and where interactions with their child do not reinforce their role as a parent or parents perceive their children as the dysfunctional (negative) element in their lives. There are 12 items in total, 11 of which are scored on a five-point Likert scale ranging from ‘strongly agree’ to ‘strongly disagree’ and 1 of which is scored on a five-point self-assessment as a parent.Difficult Child (DC): This subscale assesses some of the basic characteristics of the child that make everyday life easier or more difficult to manage. These characteristics are related to the child’s temperament, but they can also be learned as patterns of challenging or demanding behaviour. There are 12 items in total, 10 are scored on a five-point Likert scale ranging from ‘strongly agree’ to ‘strongly disagree’, and 2 are scored on a five-point scale to describe different levels of difficulty in doing or stopping something.

Although Abidin [22] (p. 62) describes the relationship between the three subscales of the PSI-SF and some of the 13 subscales of the full form, the structure of each format was kept independent. In our case, we applied the PSI-4 LF once and extracted the items corresponding to the PSI-4 SF. We obtained the scores for both scales using a single application.

#### 2.3.2. Depression, Anxiety, and Stress Scale (DASS-21)

The DASS-21 is a shortened version of the full 42-item DASS self-report scale [45]. It measures negative emotional states (anxiety, stress, and depression) and comprises seven items selected from each construct using a four-point Likert scale (corrected from 0 to 3, not applicable to very much or most of the time). A validated Spanish version of the DASS-21 exists, which has demonstrated adequate psychometric properties regarding the general adult population [46], university students [47,48], and clinical populations [49]. For this study, the Spanish version of the DASS-21 self-report instrument was used [47].

#### 2.3.3. Parental Stress Scale (PSS)

The Parenting Stress Scale (PSS) is an 18-item questionnaire that evaluates parents’ perceptions of their parenting role, considering both positive (e.g., emotional benefits and personal growth) and negative (e.g., resource demands and stress levels) aspects, using a five-point Likert scale (corrected from 0 to 4, never to very often). Developed by Berry and Jones [50], the PSS may be useful for evaluating the effectiveness of interventions designed to support the parenting skills of mothers, fathers, and/or caregivers of children of all ages. In this study, the Spanish version of the PSS [51] was used.

#### 2.3.4. Parent Behaviour Inventory (PBI)

The Parent Behaviour Inventory (PBI) [52] is a questionnaire that assesses a wide range of parenting behaviours in two broad dimensions: hostile/coercive and supportive/committed. This 20-item inventory asks parents to rate the extent to which each statement reflects their relationship with their child. Responses are given on a Likert scale ranging from 0 (not at all true) to 5 (very true). The Spanish version of the PBI [53] was used.

### 2.4. Data Analysis

For the purpose of conducting the requisite psychometric analyses, this study adhered to the guidelines established by Nunnally [54] and Raykov & Marcoulides [55].

#### 2.4.1. Descriptive and Differential Analysis

Descriptive analyses were performed on the 13 subscales, the two domain subtotals, and the total PSI-4 score, using mean and standard deviation calculations. A statistical comparison of the two groups was performed using Student’s t-test to determine the significance of the differences, and Cohen’s d was used to determine the effect size. When there were more than two groups, a single-factor ANOVA was used to determine the significance of the differences, and eta squared was used to determine the effect size [56].

#### 2.4.2. Internal Consistency

Conventionally, the Cronbach alpha coefficient [57] is used as an indicator of the internal consistency of a psychometric measurement instrument. Although this coefficient is possibly the most popular and widely used, numerous studies have been conducted questioning its appropriateness in all cases [58,59,60]. The use of this indicator is not questionable when the principles of unidimensionality or non-correlation between item measurement errors [61] or normality of scale score distribution and item metrics [54] are met.

The metric characteristics of the PSI-4 items (polytomous items) force us to reflect on what would be the best measure of internal consistency. In this regard, when comparing various internal consistency indicators based on the metric characteristics of the scale, we concluded that some alternative internal consistency tests, such as tau and omega, converge with alpha. However, in agreement with Dunn et al. [58], we believe that the omega coefficient is particularly appropriate for instruments constituting polytomous items used in social and health science research, as it is more robust, especially when these address multidimensional constructs [62].

#### 2.4.3. Test–Retest Reliability

Test–retest reliability attempts to assess the stability of the measurement over time. It is an indicator of the quality of the measurement instrument, but its accuracy is influenced by the nature of the construct being assessed. In this sense, we understand that parental stress is an emotion that can fluctuate over time; therefore, the retest cannot be delayed too long, as the characteristics or circumstances of the interaction between the child and parents may change. Typically, the difference in means between the two assessments (test and retest) and the correlation between the two measures are calculated [63].

#### 2.4.4. Construct Validity

Construct validity is the ability of an instrument to accurately measure the construct it is designed to measure [64]. This can be determined using either convergent validity or by comparing it with known clinical groups. Convergent validity refers to the degree to which scores obtained using the instrument under evaluation correlate with those obtained using other reference instruments. Regarding recognised groups, it refers to the instrument’s ability to characterise individuals from different groups [63].

#### 2.4.5. Content Validity

The aim of content validity is to examine whether the concepts of interest in the construct(s) evaluated by the instrument are adequately represented by the questionnaire’s items [65]. Content validity is analysed based on information provided by the authors regarding the construction process, item selection, and determination of the target population [63]. When a questionnaire is translated and adapted to a different language and cultural setting, it is usually designed to replicate the original’s theoretical model.

One of the usual techniques for analysing whether questionnaire items replicate the theoretical model is factor analysis (confirmatory or exploratory, as appropriate) [66]. Model fit assessment usually includes both absolute and relative fit indices. Model fit depends on both the complexity of the model and the size and type of the sample [67]. For categorical data, such as Likert scales, the WLSMV (Weighted Least Squares Mean and Variance adjusted) estimator was used [68].

In the literature, the chi-square test is used as a measure of overall model fit. In particular, a chi-square/dfratio of less than 5 is accepted as a good fit [67,69]. However, this indicator is insufficient, as it is influenced by the sample size [70]. The RMSEA index (Root Mean Square Error of Approximation) is a model fit index that works independently of the number of factors, although it is affected by the amplitude of the measurement scale. In short, it provides information on the significance of the residual variance not explained by the model.

Among the most commonly used relative indices to determine model fit, the Tucker–Lewis index (TLI) stands out, indicating the quality of the representation of the intercorrelations between the items or attributes of a scale or battery, as well as the comparative fit index (CFI). This indicator results from comparing a random model, in which there are no relationships between the variables studied, and the model proposed by the researcher. Finally, SRMR (Standardised Root Mean Square Residual) is a complementary indicator to RMSEA and, as an absolute measure of fit, it is defined as the normalisation of the difference between the observed correlation and that predicted by the model.

#### 2.4.6. Software

Version 29 of SPSS, licensed by the University of Valencia (Valencia, Spain), was used for conventional statistical calculations. Confirmatory Factor Analyses (CFAs) and Exploratory Factor Analyses (EFA) were performed using Mplus 8.3 [71].

## 3. Results

### 3.1. Descriptive Analysis

Table 2 presents the means and standard deviations of the 13 subscales of the PSI-4, its two domains (children and parents), and the total stress score. When comparing the mean values of the subscales obtained from the Spanish sample with those obtained by Abidin [22], differences with a low level of effect (Cohen’s d) are observed, which justifies the need for updated scales adapted to the cultural contexts of the reference populations.

### 3.2. Internal Consistency and Test–Retest Stability

Table 3 presents the Cronbach alpha scores for the total test, each domain (children and parents), and each of the 13 subscales. It is worth noting that the total coefficients are very satisfactory (over 0.90). Still, when analysing the behaviour of the subscales, some show lower stability scores (due to, among other reasons, the number of items in each subscale). We highlight the AD, RE, and MO scores of the Child Domain, while internal consistency in the subscales of the parent domain reaches higher values, except for the AT subscale.

On the other hand, regarding test–retest reliability, the second administration (retest) was conducted two to three months after the first. We understand that parental stress scores generated by raising a particular child should not differ significantly over short periods of time, provided there is no external intervention. The sample of typically developing children was asked to participate in the retest. As shown in Table 4, the correlation coefficients between the two measures exhibit high values, indicating that the Spanish version of the PSI-4 yields stable results over time.

### 3.3. Construct Validity

#### 3.3.1. Convergent Validity

The problem in determining convergent validity lies in establishing the procedure and selecting the gold standard for comparison. In short, it involves selecting a conventionally accepted measure of the trait and correlating it with the results of the instrument to be validated. The studies reviewed used a variety of instruments to validate the measurement of parental stress using the PSI-4 or to attempt to explain the phenomenon [6,20,21]. Three tests were selected to verify the convergent validity of the Spanish version of the PSI-4: the PSS, the DASS-21, and the PBI (described in the Section 2.3). From a statistical perspective, as the subsample in which the concurrent measures were applied was smaller (N = 87), we must treat the results with caution.

Table 4 shows the correlations between the total scores of the three selected tests and the PSI-4 subscales. It is noteworthy that the PSS score is significantly correlated with all PSI-4 subdomains. Regarding the DASS-21 subscales, higher and more significant correlations are observed with the stress subscale, although high correlations are also observed with anxiety and depression. Concerning the hostile/coercive and supportive/committed dimensions of the PBI, the stress scores correlate positively with hostile/coercive (HC) styles and negatively with supportive/committed (CS) styles. This suggests that high levels of parental stress are somewhat related to HC styles, while low levels of parental stress are related to CS styles. The measures of the different questionnaires seem to converge with the construct of the Spanish version of the PSI-4 parental stress scale.

#### 3.3.2. Known Clinical Groups

The diagnostic capacity of a psychological instrument is typically measured by its accuracy in identifying subjects with a particular disorder or disturbance that the instrument is designed to assess. In our case, we defined a group of parents whose children had a history of special educational needs or, due to their age, were attending an early childhood intervention centre. For comparison, a parallel group was randomly selected from the rest of the sample, consisting of two groups balanced in age and gender, comprising 65 children. Table 5 presents the results of the comparison between the two groups, highlighting that the differences between them are all statistically significant, except for the AT (Attachment) subdomain. At this point, we can conclude, based on the specialised literature [32,72,73], that parents of children with special educational needs experience higher levels of parental stress, and that the Spanish version of the PSI-4 is capable of assessing this.

### 3.4. Content Validity

Since its origin, the theoretical model of the PSI has been analysed with varying degrees of success [22]. Content validity attempts to measure the relevance of the scores obtained with the instruments regarding the psychological concepts they are intended to measure. In the Spanish adaptation of the PSI-4, we aimed to demonstrate that neither translation nor adaptation to cultural norms introduces variations into the scale’s structure, and therefore, the original model remains intact. In the construction of the PSI [22,30], two principal component analyses (PCAs) were conducted (one for each of the domains). A PCA was conducted on the 47 items belonging to the Child Domain, which, with six factors, explained 53% of the variance. According to Abidin [22] (p. 42), all factor loadings improved compared with the PSI-3 factor analysis, with the exception of Demandingness (DE). The items in this subscale were more consistently distributed across the Acceptability (AC) and Mood (MO) subscales, consistent with previous studies [30]. The nine items comprising DE have face validity with the construct; therefore, clinical interpretation was maintained [22]. A second PCA was calculated on the 54 items of the Parent Domain, which, with seven factors, explained 54% of the variance. Overall, the factor loadings of the subscales of both domains improved compared with those in the PSI-3. In the Parent Domain, the items of the Health (HE) subscale did not appear as an independent factor and were associated with the Depression (DP) subscale. Although the data did not demonstrate the existence of this scale as an independent factor, it was maintained as such due to its clinical implications. The final factor analysis of the 13 subscales also yielded favourable results; two factors explaining 72% of the variance were specified with higher factor loadings than those on PSI-3 [22] (p. 45). It is important to consider that according to the PSI theoretical model, all sources of stress accumulate and contribute to the final score. Consequently, the subscales should provide meaningful clinical information, even if some of them do not behave independently in factor analysis.

The theoretical model to be tested consists of organising the 101 items into 13 subdomains as first-order factors and two domains as second-order factors. The estimation method used was WLSMV (Weighted Least Squares Mean and Variance adjusted), implemented in Mplus 8.3 [71], which is the most recommended approach for categorical data, such as Likert scales [74,75].

Table 6 presents the fit indicators used and the values obtained for verifying the model, which comprises 13 first-order factors and 2 second-order factors. The Chi-square, RMSEA, and SRMR values are close to the critical acceptance values; however, the CFI and TLI indices are not acceptable.

Since the original structure was not fully confirmed, an EFA was performed on the same sample to explore a possible different dimensionality [76]. We followed the PSI-4 construction philosophy and attempted to test the model on each sub-domain independently. However, we used EFA with WLSMV estimation and oblimin rotation, as implemented in Mplus 8.3 [71].

Table 7 presents the fit indices for the six-factor solution for the Child Domain and the seven-factor solution for the Parent Domain. For the Child Domain (see Table 8), an EFA was performed on the 47 items. A maximum of six factors were required (corresponding to the number of subscales that the PSI-4 presents in the Child Domain). All of them have eigenvalues greater than 1.0. The first factor comprises items from the DI (Distractibility/Hyperactivity) and DE (Demandingness) scales; it expresses, to some extent, parent–child interaction difficulties. The second factor mainly comprises items from the AC (Acceptability), DI (Distractibility/Hyperactivity), and DE (Demandingness) scales. The third factor comprises RE (Reinforces Parent) and DE (Demandingness) scale items. The fourth factor is defined by items from the MO (Mood) and AD (Adaptability) scales. The fifth factor comprises items from AC (Acceptability), AD (Adaptability), and DE (Demandingness). Finally, the sixth factor combines four items from four different scales (RE, MO, AD, and DE). It should be noted that 10 items do not show significant loadings in any of the first six factors. In conclusion, the RE, DI, MO, AC, and AD scales are defined by the factors obtained; however, there are items that saturate several factors, demonstrating the inter-relationships observed between the different scales that make up the Child Domain. Particularly noteworthy are the items on the AC (Acceptability) scale and the items on the DE (Demandingness) scale, which are present in up to four factors. Table 9 shows the correlations between the factors.

For the parenting domain (see Table 10), an EFA of the 54 items was performed. A maximum of seven factors were requested (corresponding to the number of subscales that the PSI-4 presents in the Parent Domain). All of them have eigenvalues greater than 1.0.

The first factor is defined by three items from the CO (Competence) scale; the second factor comprises items from the CO (competence) and AT (Attachment) scales; the third factor constitutes items from the RO (Role Restriction) scale; the fourth factor is formed by items from the DP (Depression) scale; the fifth factor comprises six items from the SP (Spouse/Parenting Partner Relationship) scale; the sixth factor constitutes IS (Isolation) scale and HE (Health) scale items; finally, the seventh factor is defined by only two items from the DP (Depression) scale. As in the case of the Child Domain, 10 items do not show significant saturations in any of the seven factors. Although there is less evidence than in the case of the Child Domain, there are items that saturate more than one factor. Table 11 shows the correlations between the seven factors in the Parent Domain.

Recall that in the original Abidin study [22], the factor structure obtained from the items was also fuzzy. As in the original study, we tested the model using the 13 subscale scores organised into two first-order factors and one total second-order factor (see Figure 2). In this case, the fit indices improved (see Table 12), enabling us to accept the overall structure of the PSI-4. We also tested the three-factor structure, dividing the Parent Domain into two factors, as shown in Figure 1. Note that Abidin’s original model [22] suggests that the Parent Domain can be organised into two dimensions: situational and personal factors. The three-factor structure (see Table 12 and Figure 3) has better fit indicators than the two-factor model.

### 3.5. Short Form PSI-4

The short version of the PSI was obtained by selecting items from the long form using various factor analyses [22]. The 36 items are structured into three scales, each with the same number of items. The three scales were labelled Parental Distress (PD), Parent–Child Dysfunctional Interaction (P-CDI), and Difficult Child (DC). Table 13 presents the psychometric indicators of the PSI-4 SF. As can be seen, the values for internal consistency and measurement stability follow the same pattern as in the long version of the instrument.

Regarding the factor structure of the PSI-4 SF, a CFA was performed [71]. In this case, the model to be tested is based on the 36 items, organised into three dimensions or first-order factors and a total second-order factor. As shown in Table 14 and Figure 4, the results demonstrate a better fit than in the case of the PSI LF (see Table 10). However, the fit between the empirical data obtained from the non-clinical sample and the theoretical model of the PSI-4 SF can only be considered adequate overall.

## 4. Discussion

Evidence from the specialised literature suggests that the construct evaluated (parental stress) is an important factor in the psychological health and well-being of parents, children, and the family as a whole [18,79,80]. Parental stress often emerges at various stages during child-rearing and sometimes remains at high levels for prolonged periods. To plan possible interventions from the psychologist’s or paediatrician’s office, it is necessary to have screening and assessment tools translated into various languages and adapted to the cultural context of each country. The differences in mean scores between the Spanish and American samples alone justify the need for adaptation, demonstrating how culture can influence the evaluation of parenting behaviours and customs. The PSI is the international standard for assessing parental stress; however, its psychometric performance in non-clinical Spanish samples remains unknown. Although there are Spanish versions of the PSI-4, particularly the abbreviated form for research purposes, to date, there has been no official version with specific norms tailored to the Spanish population and its cultural context. The aim of this study was to address this issue.

Comparing the average scores obtained in our study with those in Abidin’s original study [22] requires the development of specific norms for the Spanish population. Furthermore, differences in average parental stress scores according to children’s age necessitate the presentation of different norms [13,14]. No differences were found according to the child’s gender, and the available data do not allow us to analyse differences between fathers and mothers or parents’ partners.

The Spanish version of the PSI-4 has demonstrated strong psychometric properties, similar to those of the original version or those obtained in other studies, indicating its suitability for measuring parental stress in a valid and reliable manner across cultures. The internal consistency indices can be considered excellent and even too high (above 0.90). This suggests that there may be some redundant items, and as a result, the scale may be longer than necessary [81]. These data justify the demand for a shorter form from clinicians and researchers. However, the long form provides relevant clinical information for planning an intervention, if necessary.

We understand that high parental stress scores can remain stable over time in the absence of intervention. Although the group of parents who participated in the retest in our study was small, high stability was observed in the total score, both in the short and long forms. In this sense, our study provides valuable information rarely found in other studies. The review by Holly et al. [6] highlights this gap in published psychometric studies.

Construct validity cannot be determined by a single analysis. Two approaches have been adopted to determine this: convergent validity, which utilises other measures of the same construct or other constructs theoretically consistent with the measure used, and discriminant validity, which is the ability of the measure to differentiate between groups with clinical symptoms and other groups. Specifically, the PSI-4 in all its forms and subscales has been found to correlate with the PSS score. The present study also demonstrates a correlation with a measure of stress and, to a lesser extent, with general anxiety and depression, measured using the DASS-21. The PSI-4 subscales show a high degree of correlation with PBI scores. Regarding discriminatory power, positive results have been obtained when comparing groups of parents of typically developing children with those parents whose children have a developmental disorder requiring intervention.

Studies reviewing the content validity of the PSI have yielded positive findings [6,20]. The items comprising the PSI were selected from the literature on child development, parent–child interaction, parenting practices, and child psychopathology. This aspect has been improved through the pilot test and subsequent versions of the PSI. However, the factorial structure of the PSI has been the subject of numerous studies [22], particularly those involving translated and adapted versions in other languages. Notable among these are the adaptations into Turkish and Portuguese for the Brazilian population.

In the Turkish adaptation of the instrument [82], a CFA was used to verify the scale structure following the methodology used by the PSI author; the structure was validated by two confirmatory factor analyses on a sample of 386 parents, one for the Child Domain (CD) and another for the Parent Domain (PD). The results of the CFI and TLI fit indices were below the cut-off point used in our study; however, they accepted that the structure of each domain was consistent with the author’s model.

In the validation of the Portuguese version in Brazil [36], a different technique was used. Factor analysis (principal components, with varimax rotation) was performed on the scores of the thirteen subdomains obtained from a sample of 53 mothers of preterm infants, yielding a bifactorial structure that explained 64.57% of the variance. As this analysis is exploratory in nature, no measures of fit are available. However, the authors interpret the results as those of the two original domains (CD and PD), although they report that in the Child Domain, subscales such as Competence (CO) and Depression (DP) are also significantly saturated.

In essence, the PSI was constructed according to the philosophy of a construct composed of several independent subconstructs. However, the reality is not, in fact, orthogonal. A close examination of the components that constitute CD and PD reveals a clear obliquity between them. This obliquity is further accentuated when the items within each subscale are analysed.

On the other hand, in addition to the cultural differences that may exist in the way parental relationships are understood, there have been social and cultural changes in the last 20 years since the original PSI-4 version was published. We can justify the existence of certain differences in the factor structure of the theoretical model proposed by Abidin, the one obtained empirically in the PSI-4 typing studies, and the current ones. However, given the clinical significance of the PSI-4 subscales, we propose maintaining the same structure to facilitate comparisons in cross-cultural and cross-population studies.

The aim of the present work was to study the fourth version, both in its long and short forms—given the interest in having a version with fewer items—and with a normative population. The short form can be used for screening or in clinical contexts, while the long form can be used to obtain broader and deeper information when needed.

It is important to note that most of the studies cited used clinical samples composed of parents at risk, whereas our study was conducted with parents of children without known developmental disorders. This difference may be significant when analysing latent variables in a factor analysis. Nevertheless, the two-factor model, built on the subscale scores, is reproduced in the same way, improving the fit with respect to a three-factor model where the Parent Domain is subdivided into two factors (personal and situational). In the short form version, the fit to the original three-factor model is more adequate; perhaps this is one of the reasons, along with its ease of interpretation and application, that the short form is much more widely used than the long form. In our opinion, both forms are complementary: the PSI-4 SF can be used as a screening tool, and in cases where high scores are observed in any of the three factors (especially in the P-CDI factor), obtaining the complete PSI-4 profile can help us better plan a possible intervention. Furthermore, it is clinically useful, as research has shown that parental stress is higher in parents of children with a clinical diagnosis, as reported in this study. Parental stress can be related to poorer treatment outcomes, reducing their effectiveness due to lack of adherence and a decrease in the quantity and quality of interactions with children in daily activities. Parental stress monitoring allows clinicians to adjust treatment and identify risks for its continuation; therefore, the instrument must be sensitive to changes in parental stress throughout treatment.

## 5. Conclusions

The Spanish version of the PSI-4, in its two forms (LF and SF), demonstrated good psychometric qualities, indicating that it is capable of reliably and validly measuring parental stress in the Spanish population. The differences between the mean scores of the American and Spanish samples indicate the need to generate differential scoring norms.

The internal consistency of the PSI-4 in both versions is very acceptable, as is its test–retest stability. Regarding convergent validity and ability to differentiate between clinical groups, the PSI-4 is a tool that can be used for clinical purposes (screening and diagnosis) and to monitor treatment follow-up. Convergent validity with other consistent constructs indicates that the Spanish version of the PSI-4 assesses the construct of parental stress. Furthermore, a comparison between parents with typically developing children and those with children who require support or intervention demonstrates discriminative ability.

Clearly, further research is required to develop a theoretical model that incorporates mediating relationships between its components.

### 5.1. Limitations

Given the sample structure (characterised by a low proportion of fathers) and the medium to high educational level of the respondents, it is possible that the diversity of family structures in Spain is not fully represented. The need for respondents to have a good level of reading comprehension should be taken into account.

Furthermore, the relatively small size of the subsamples used to determine convergent validity and test–retest reliability could affect the value of the correlations, overestimating them. The convergent validity analysis was based solely on a subsample of 87 parents, resulting in a small sample size for test–retest reliability, which could lead to overestimating the stability of the correlations.

The timing of the application (weeks before the lockdown due to the COVID-19 pandemic) and the pause from March 2020 to January 2021 may have introduced some bias that should be taken into account.

### 5.2. Future Lines of Research

Regarding new lines of research, verification of the factorial structure of the PSI-4 LF requires the development of new studies. For a more comprehensive understanding of the PSI, it is essential to examine a new model that incorporates oblique relationships or mediation functions between the subscales. This model will facilitate the identification of advantages or disadvantages from a clinical perspective, thereby enhancing our ability to predict and diagnose psychological conditions.

Furthermore, although self-reports have demonstrated robust psychometric properties, the potential of a mixed-methods assessment strategy warrants further investigation. Currently, most studies with clinical samples rely on self-reporting; using observational, physiological, or performance assessment methods could help us better understand how parents experience parenting stress.

There is considerable heterogeneity in parenting experiences among parents, depending on their background, culture, educational level, and gender. Therefore, it is necessary to refine and detail the psychometric performance of the measure among different groups of parents (mothers versus fathers, parents by gender, low socioeconomic status, and single-parent families, among others).

## Figures and Tables

**Figure 1 children-12-01466-f001:**
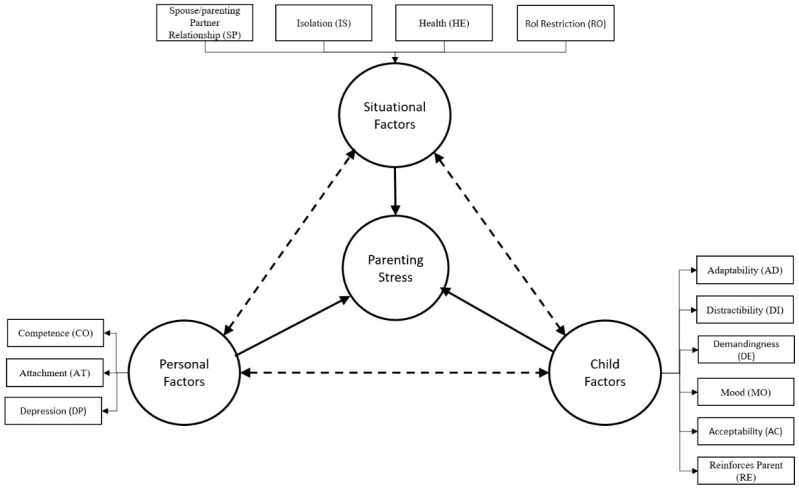
Model of Parenting stress adapted and modified from Abidin [1,12,18].

**Figure 2 children-12-01466-f002:**
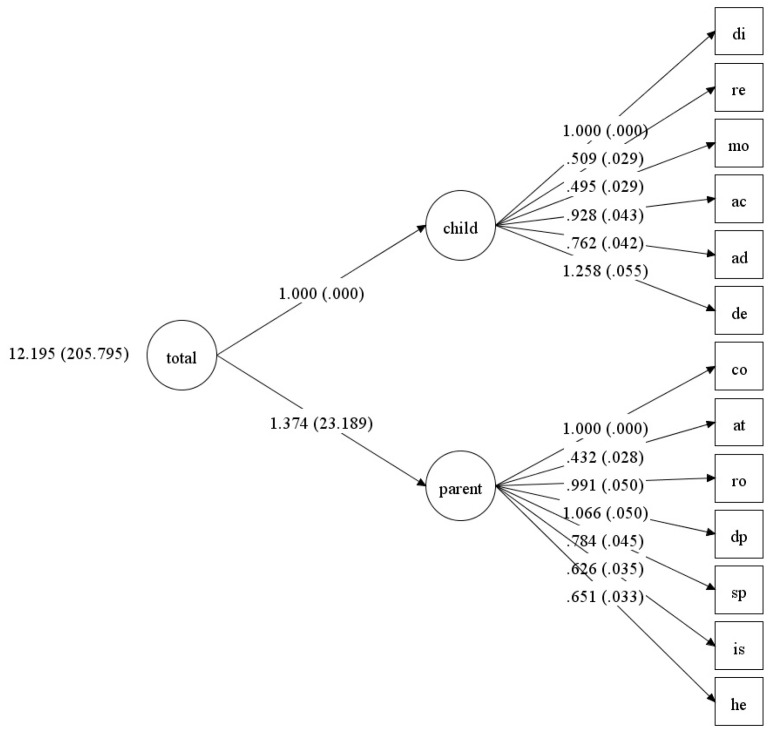
Diagram of the 13-value model organised into two large first-order factors and a single second-order factor (standard error values are shown).

**Figure 3 children-12-01466-f003:**
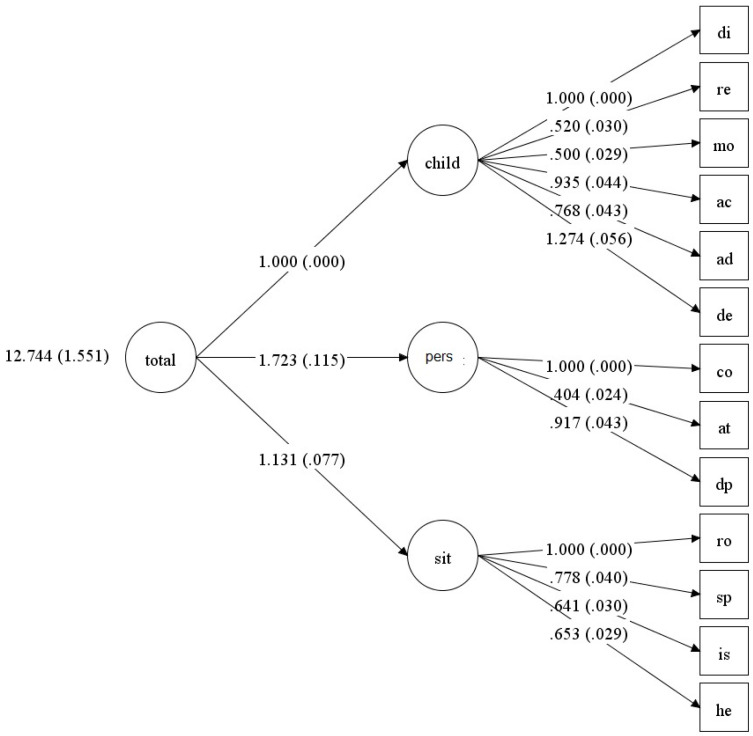
Diagram of the 13-value model organised into three large first-order factors and a single second-order factor (standard error values are shown).

**Figure 4 children-12-01466-f004:**
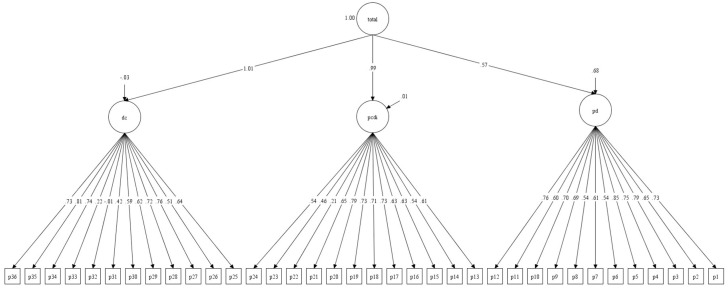
Diagram showing the relationships between the 36 items of the PSI-4 SF and the three-factor structure grouped into a single second-order total factor.

**Table 1 children-12-01466-t001:** Sample characteristics.

Variable	Mean	(Std) *	% Sample
Parent			
Mother	40	(4.6)	82.1
Fathers	42	(5.4)	17.9
Child gender			
Male			54.5
Female			45.5
Education level: mother			
Primary school			13.0
Secondary school			33.2
High school/bachelor’s			45.4
College graduate or higher			8.4
Education level: father			
Primary school			15.6
Secondary school			27.3
High school/bachelor’s			48.4
College graduate or higher			8.7

* Std Standard Deviation.

**Table 2 children-12-01466-t002:** Means and standard deviations of PSI-4 subdomains in the Abidin [22] typing sample and the Spanish typing sample.

	Spanish Sample		Abidin Sample [22]				
	Mean	Std	Mean	Std	t	d Cohen	*p*
Distractibility/Hyperactivity (DI)	25.24	7.07	22.5	6.4	8.70	0.41	>0.01
Reinforces Parent (RE)	11.56	3.98	11.4	4.4	0.83	0.04	<0.05
Mood (MO)	12.85	3.79	11.5	3.9	7.56	0.36	>0.01
Acceptability (AC)	12.95	5.53	13.4	5.6	−1.74	−0.08	>0.05
Adaptability (AD)	25.31	5.54	24.8	7.3	1.72	0.08	>0.05
Demandingness (DE)	19.16	6.95	19.7	6.7	−1.70	−0.08	>0.05
Child Domain	107.09	25.92	103.2	29.5	3.04	0.14	>0.01
Competence (CO)	31.34	7.26	29.5	8.4	5.09	0.24	>0.01
Attachment (AT)	10.99	3.75	13.0	5.3	−9.63	−0.46	>0.01
Role Restriction (RO)	20.19	6.44	17.8	5.4	8.58	0.40	>0.01
Depression (DP)	19.68	6.69	19.1	7.0	1.83	0.09	>0.05
Spouse/Parenting Partner Relationship (SP)	16.85	5.97	17.5	6.6	−2.24	−0.11	>0.01
Isolation (IS)	13.54	4.58	13.3	4.7	1.12	0.05	<0.05
Health (HE)	12.06	4.31	11.1	3.9	5.00	0.23	>0.01
Parent Domain	124.68	29.50	121.4	34.3	2.23	0.10	>0.01
Total Stress	231.78	50.10	224.6	60.7	2.81	0.13	>0.01
N	828		1056				

**Table 3 children-12-01466-t003:** Coefficient of internal consistency (alpha and omega) and test–retest values of measurement stability.

	Number of Items	Alpha	Omega	Test–Retest
Distractibility/Hyperactivity (DI)	9	0.85	0.89	0.91
Reinforces Parent (RE)	6	0.79	0.79	0.73
Mood (MO)	5	0.76	0.83	0.82
Acceptability (AC)	7	0.90	0.93	0.65
Adaptability (AD)	11	0.73	0.84	0.76
Demandingness (DE)	9	0.89	0.90	0.79
Child Domain	47	0.97	0.98	0.87
Competence (CO)	13	0.80	0.82	0.87
Attachment (AT)	7	0.82	0.85	0.79
Role Restriction (RO)	7	0.90	0.91	0.81
Depression (DP)	9	0.91	0.91	0.90
Spouse/Parenting Relationship (SP)	7	0.87	0.87	0.82
Isolation (IS)	6	0.84	0.85	0.82
Health (HE)	5	0.83	0.83	0.84
Parent Domain	54	0.97	0.99	0.93
Total Stress	101	0.99	0.99	0.92
N	828			25

Internal consistency coefficient was calculated from standardised factor coefficients (CFA) according to Viladrich, Angulo-Brunet, and Doval [62]. Test–retest value of measurement stability was calculated as the Pearson correlation coefficient between the two measures.

**Table 4 children-12-01466-t004:** Correlations between PSI-4 subdomains and total scores, as well as the criteria questionnaires used.

	PSS	DASS-21			PBI	
		Stress	Anxiety	Depression	HC	CS
Distractibility/Hyperactivity (DI)	0.50 **	0.42 **	0.31 **	0.28 **	0.46 **	−0.32 **
Reinforces Parent (RE)	0.21 *	0.40 **	0.27 *	0.36 **	0.40 **	−0.50 **
Mood (MO)	0.33 **	0.38 **	0.28 **	0.28 **	0.38 **	−0.14
Acceptability (AC)	0.23 *	0.23 *	0.19	0.19	0.23 *	−0.47 **
Adaptability (AD)	0.30 **	0.35 **	0.33 **	0.30 **	0.25 *	−0.19
Demandingness (DE)	0.34 **	0.55 **	0.49 **	0.50 **	0.48 **	−0.35 **
Child Domain	0.42 **	0.50 **	0.42 **	0.41 **	0.47 **	−0.41 **
Competence (CO)	0.43 **	0.63 **	0.51 **	0.51 **	0.47 **	−0.31 **
Attachment (AT)	0.27 *	0.44 **	0.30 **	0.44 **	0.39 **	−0.51 **
Role Restriction (RO)	0.50 **	0.65 **	0.47 **	0.54 **	0.56 **	−0.17
Depression (DP)	0.49 **	0.71 **	0.50 **	0.55 **	0.50 **	−0.24 *
Spouse/Parenting Relationship (SP)	0.46 **	0.44 **	0.28 **	0.38 **	0.41 **	−0.28 **
Isolation (IS)	0.39 **	0.47 **	0.37 **	0.54 **	0.23 *	−0.24 *
Health (HE)	0.36 **	0.71 **	0.62 **	0.63 **	0.45 **	−0.26 *
Parent Domain	0.54 **	0.75 **	0.56 **	0.65 **	0.56 **	−0.35 **
Total Stress	0.52 **	0.68 **	0.53 **	0.58 **	0.56 **	−0.40 **

** Significant correlation at 0.01; * significant correlation at 0.05.

**Table 5 children-12-01466-t005:** Mean values of the PSI-4 subscales and totals, and F ratio of differences between groups of families with typically developing children and families with children with atypical development.

	Typical Development		Clinical Sample			
	Media	Std	Media	Std	F	*p*
Distractibility/Hyperactivity (DI)	25.37	6.67	32.72	7.68	33.93	<0.001
Reinforces Parent (RE)	9.88	2.88	15.89	3.92	99.21	<0.001
Mood (MO)	13.28	3.85	15.29	3.92	8.71	0.004
Acceptability (AC)	11.42	4.27	21.55	6.65	106.74	<0.001
Adaptability (AD)	24.32	5.30	31.02	4.90	55.76	<0.001
Demandingness (DE)	17.57	5.92	29.86	7.20	112.89	<0.001
Child Domain	101.83	21.39	146.34	28.40	101.83	<0.001
Competence (CO)	28.71	5.88	39.14	5.00	118.57	<0.001
Attachment (AT)	10.38	3.01	12.40	4.11	10.12	0.002
Role Restriction (RO)	19.77	6.00	24.20	7.25	14.40	<0.001
Depression (DP)	18.72	5.76	22.82	7.35	12.47	<0.001
Spouse/Parenting Relationship (SP)	15.55	4.68	19.42	6.30	15.71	<0.001
Isolation (IS)	13.09	3.84	16.14	4.92	15.43	<0.001
Health (HE)	10.92	4.14	14.40	5.12	18.09	<0.001
Parent Domain	117.15	24.23	148.51	30.93	41.38	<0.001
Total Stress	218.98	40.54	294.85	54.19	81.66	<0.001
N	65		65			

**Table 6 children-12-01466-t006:** Indicators of the fit to the 101-item theoretical model in 13 first-order factors and two second-order factors.

	Cut-Off Point	CFA PSI-4
Chi-square test of model fit		20,767.38
df		4935
*p*-value		0.00
RMSEA	<0.08	0.062
90% confidence interval	<0.10	0.061–0.063
CFI	0.90–0.95	0.764
TLI	0.90–0.95	0.759
SRMR	<0.08	0.081

Cut-off points obtained from [76,77,78].

**Table 7 children-12-01466-t007:** Indicators of the fit to the two domains with EFA WLSMV. Six factors for the Child Domain and seven for the Parent Domain, according to the original PSI-4 distribution.

	Cut-Off Point	EFA Child Domain S	EFA Parent Domain
Chi-square test of model fit		2154.91	3144.22
df		814	1074
*p*-value		*0.00*	*0.00*
RMSEA	<0.08	0.047	*0.051*
90% confidence interval	<0.10	0.045–0.049	0.049–0.053
CFI	0.90–0.95	0.959	0.943
TLI	0.90–0.95	0.946	0.924
SRMR	<0.08	0.037	0.041

Cut-off points obtained from [76,77,78].

**Table 8 children-12-01466-t008:** Results of the first six factors (EFA estimator WLSMV with oblimin rotation) for the Child Domain (to facilitate reading of the table, factor coefficients below 0.50 have been omitted).

Subscales	Item Number	Factors					
		1	2	3	4	5	6
Distractibility/Hyperactivity (DI)	1	0.54					
	2	0.83					
	3		0.80				
	4		0.87				
	5						
	6	0.56					
	7	0.75					
	8	0.65					
	9	0.57	0.82				
Reinforces Parent (RE)	10			0.58			
	11			0.60			
	12			0.61			
	13			0.78			
	14			0.82			
	15						0.92
Mood (MO)	16						0.53
	17				0.58		
	18				0.88		
	19						
	20				0.90		
Acceptability (AC)	21		0.72			0.58	
	22		0.73	0.44		0.56	
	23		0.86			0.60	
	24		0.55	0.69		0.62	
	25		0.58			0.71	
	26		0.79	0.56			
	27						
Adaptability (AD)	31						
	32				0.69		
	33						
	34			0.58		0.58	
	35					0.59	
	36						
	37					0.57	
	38					0.81	
	39					0.54	
	40						0.96
	41						
Demandingness (DE)	42						0.90
	43					0.58	
	44	0.55	0.67	0.52		0.62	
	45						
	46						
	47	0.63	0.61	0.63		0.58	
	48	0.65	0.64	0.56		0.62	
	49	0.62	0.66	0.60		0.66	
	50	0.55	0.59			0.59	

**Table 9 children-12-01466-t009:** Correlation oblimin factors of the Child Domain (* significant at the 5% level).

	1	2	3	4	5	6
1	1.00					
2	0.44 *	1.00				
3	0.25 *	0.44 *	1.00			
4	0.46 *	0.38 *	0.30 *	1.00		
5	0.21 *	0.48 *	0.40 *	0.26 *	1.00	
6	0.12 *	0.21 *	0.10 *	0.11 *	0.11 *	1.00

**Table 10 children-12-01466-t010:** Results of the first seven factors (EFA estimator WLSMV with oblimin rotation) for the Parent Domain (to facilitate reading of the table, factor coefficients below 0.50 have been omitted).

Subscales	Item Number	Factors						
		1	2	3	4	5	6	7
Competence (CO)	28		0.51		0.65			
	29			0.51				
	30		0.52					
	51		0.51		0.51			
	52				0.50	0.52		
	53	0.78						
	54							
	55		0.52				0.50	
	56		0.51		0.66	0.51	0.55	
	57	0.89						
	58	0.81						
	59							
	60							
Attachment (AT)	61		0.51					
	62							
	63		0.84					
	64							
	65		0.71					
	66		0.64					
	67							
Role Restriction (RO)	68			0.61				
	69			0.79				
	70			0.84		0.59		
	71			0.85		0.55		
	72			0.72			0.53	
	73			0.84			0.65	
	74			0.64				
Depression (DP)	75		0.54		0.63		0.55	
	76						0.55	
	77				0.83			
	78				0.87			
	79			0.53	0.66	0.53	0.62	
	80				0.50			
	81							0.87
	82				0.63			
	83							0.85
Spouse/Parenting Partner Relationship (SP)	84					0.63		
	85					0.84		
	86			0.52		0.76		
	87					0.72		
	88					0.63		
	89					0.66		
	90					0.54	0.50	
Isolation (IS)	91						0.82	
	92						0.80	
	93						0.70	
	94						0.82	
	95							
	96							
Health (HE)	97					0.55	0.60	
	98							
	99							
	100			0.59		0.51	0.71	
	101						0.63	

**Table 11 children-12-01466-t011:** Correlation oblimin factors for the Parent Domain (* significant at the 5% level).

	1	2	3	4	5	6	7
1	1.00						
2	0.09 *	1.00					
3	0.08 *	0.22 *	1.00				
4	0.09 *	0.27 *	0.36 *	1.00			
5	0.07 *	0.26 *	0.48 *	0.28 *	1.00		
6	0.11 *	0.340 *	0.48 *	0.42 *	0.43 *	1.00	
7	0.01	0.18 *	0.12 *	0.18 *	0.22 *	0.26 *	1.00

**Table 12 children-12-01466-t012:** CFA results for the structure of two first-order and one second-order factors on the 13 PSI-4 subscale scores.

	Cut-Off Point	Two Factors	Three Factors
Chi-square test		545.934	389.463
df		63	62
*p*		0.000	0.000
RMSEA	<0.08	0.096	0.080
90% confidence interval	<0.10	0.089–0.104	0.072–0.088
CFI	0.90–0.95	0.912	0.941
TLI	0.90–0.95	0.892	0.925
SRMR	<0.08	0.065	0.046

Cut-off points obtained from [76,77,78].

**Table 13 children-12-01466-t013:** Internal consistency and test–retest reliability of the PSI-4 SF.

	Number of Items	Alpha	Omega	Test–Retest
Parental Distress (PD)	12	0.91	0.91	0.86
Parent–Child Dysfunctional Interaction (P-CDI)	12	0.87	0.88	0.84
Difficult Child (DC)	12	0.85	0.86	0.78
Total	36	0.96	0.98	0.85

**Table 14 children-12-01466-t014:** Results of the PSI-4 SF model fit indicators.

	Cut-Off Point	CFA PSI-4 SF
Chi-square test of model fit		4132.90
df		5 91
*p*-value		0.00
RMSEA	<0.08	0.085
90% confidence interval	<0.10	0.085–0.088
CFI	0.90–0.95	0.85
TLI	0.90–0.95	0.84
SRMR	<0.08	0.07

Cut-off points obtained from [76,77,78].

## Data Availability

The data used for this research are from the national scaling published by TEA Hogrefe.

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
