# Peer review of "Validity Study of the Spanish Version of the Parental Stress Index (PSI-4) in Its Two Forms (Long Form and Short Form)"

_children, 2025, doi:10.3390/children12111466_

Round 1
Reviewer 1 Report (New Reviewer)
Comments and Suggestions for Authors
I would like to express my gratitude to Children for the opportunity to collaborate and to evaluate this very interesting and high-quality work.
The article addresses a very interesting and highly relevant topic: the validation of the PSI-4 parental stress questionnaire, both in its long and short versions.
In addition to its research excellence, the work is highly valuable for its societal impact, given the great practical utility of these instruments. The tools developed are essential for generating data that enables the creation of effective intervention strategies aimed at enhancing children's development and their parents' well-being.
The article is well-founded, both theoretically and methodologically. The authors also indicate future perspectives and limitations. The writing is clear and coherent.
In order to improve the manuscript, which, as I indicated, is already of good quality, I think it would be interesting to add these two points. I believe that the second, especially, could be of great help and learning for other readers and researchers.
1) Specify the type of responses required by the DASS-21 and PSS scales. Regarding the PBI, it is indicated: Responses are given on a Likert scale ranging from 0 (not at all true) to 5 (very true). However, regarding the other instruments, the type of response is not specified. Although references are provided that readers can consult to find this information, adding it to the manuscript would improve its quality.
2) Why was the DASS-21 version (Bados et al., reference number 47) chosen for university students and not the version for the general adult population [46] or the other version also for university students [48]?
Congratulations to the authors.
Author Response
Dear reviewer, we appreciate the time you have devoted to reading our work and to providing questions and suggestions.
Regarding question 1, the suggested improvements have been implemented:
For the DASS-21, ‘using a four-point Likert scale (corrected from 0 to 3, not applicable to very often or most of the time)’.
For the PSS: ‘using a five-point Likert scale (corrected from 0 to 4, never to very often)’.
Regarding question 2,
The version by [46] Daza, P., Novy, D., Stanley, M., & Averill, P. (2002) is a study developed for Spanish speakers in the USA and contains some expressions that are not appropriate for the Spanish population. It also has a sample limitation in its application (it was only applied to 98 participants). Regarding the decision to choose between the version [47] Bados, A., Solanas, A., & Andrés, R. (2005) or the version [48] Fonseca-Pedrero, E., Paino, M., Lemos-Giráldez, L., & Muñiz, J. (2010), these are two studies, the first [47] on the construction and validation of the short version (evolution from DASS-42 to DASS-21) and the second on the validation of that same tool applied in its short form DASS-21. It is therefore the same questionnaire.
We hope we have replied to your questions and concerns, and we thank you again for your time and dedication.
Reviewer 2 Report (New Reviewer)
Comments and Suggestions for Authors
Thank you for the opportunity to review this study. This study shows strong empirical support for the psychometric properties of the PSI-4. I provide feedback to help the manuscript reach its fullest potential.
- In the introduction, it is unclear what is meant regarding that stress is the sum of all factors - what is meant by "all factors"?
- The order of the methods is difficult to understand. When did participants provide consent (and how did they provide consent)? How was consent collected and then the anonymous surveys?
- The break in data collection should be listed as a limitation of the study.
- In Table 1, what does "STD" stand for?
- In Table 1, the percentages for mothers' and fathers' education do not equal 100%.
Author Response
Thank the reviewer for the time spent reading the article. Thank them for commenting on the content.
1. In the introduction, it is unclear what is meant regarding that stress is the sum of all factors - what is meant by "all factors"?
The PSI-4 is constructed following the model of Lazarus and Folkman [7], which understands stress to be the result of the interaction between different stressors, cognitive competencies, and coping responses. Therefore, the final equation is the sum of the different components. Consequently, the model proposed by Abidin is to add up all the factors that contribute to stress in parenting. Abidin operationalises this with 13 subdomains organised between the domain of parents and the domain of children.
2. The order of the methods is difficult to understand. When did participants provide consent (and how did they provide consent)? How was consent collected and then the anonymous surveys?
Clarifications have been added to the ‘Procedure’ section regarding when the informed consent form is delivered and when it is returned signed.
3. The break in data collection should be listed as a limitation of the study.
This is mentioned in the 'Limitations' section, which states:
'The timing of the application (in the weeks leading up to the lockdown due to the Covid-19 pandemic) and the pause from March 2020 to January 2021 may have introduced some bias that should be taken into account'.
4. In Table 1, what does "STD" stand for?
STD = Standard Desviation.
5. In Table 1, the percentages for mothers' and fathers' education do not equal 100%
The mistake has been corrected. Thank you for pointing it out.
Thank you for your time and dedication.
Reviewer 3 Report (New Reviewer)
Comments and Suggestions for Authors
Journal: Children
Manuscript ID: children-3928439
Type: Article
Title: Validity Study of the Spanish Version of the Parental Stress Index (PSI-4) in its Two Forms (Long Form and Short Form)
Date of Review: 18/10/2025
- General
Many thanks for the opportunity to review your manuscript. It addresses an important issue that is very relevant to this journal. It cuts across issues of parenting, mental health, and child and family well-being. This is a very important topic that is relevant to the journal. The manuscript could benefit from minor revisions; more comments, suggestions, and guidance are provided below.
- Title
Thank you, the title is very clear and well formulated.
- Abstract
Your abstract is well-written and provides an overview of how the study was undertaken. I am not sure whether the information presented before the abstract (Lines 9-25) is necessary. More information on the methods could be beneficial, e.g. what sampling method was used, what was the research type, it is implied that it was a quantitative study, but this needs to be more explicit. What method was used to analyse data?
- Introduction
Your introduction is well written and captures all the important aspects of the paper. It is clear that you understand the subject area.
Statistics provided in lines 53-54 should be referenced.
Your review of other studies and presentation of consolidated information is impressive. Well-done. Thank you for adding Figure 1; it provides a quick visual grasp of the model of parenting stress.
Your introduction also provides a solid rationale and justification of the study, which nicely leads to the materials and methods section. Well done.
- Materials and Methods
Your methodology section is very detailed and clearly explains how the study was conducted, well done.
In line 146, you state, “As the response rate using this method was limited.” It might be worthwhile to clearly indicate the percentage of the response rate and what literature/ theory says is the acceptable response rate.
Whilst you state that the study received ethical clearance, you should also discuss the ethical considerations of this study. This information is very important, especially considering the sensitivity of the study.
- Results
Your results section is elaborate, solid, and well presented. The deliberate use of tables to provide a consolidated, quick grasp of the results is very impressive. You came out with striking results. Well done.
- Discussion
Your discussion section is well thought out and solid. It shows a clear alignment with the results of the study and is supported by an integration of relevant literature. This section clearly shows how your study findings fit in the broader discussions and how they contribute to the field. Well-done.
- Conclusions
Your conclusions are clearly articulated and are aligned with what is presented in the results and discussion sections. However, instead of presenting them in very short paragraphs, which makes it seem as if they are presented in point form, I suggest that you consolidate all these short paragraphs into one solid paragraph.
Moreover, I suggest that Heading No. 6. Future lines of research, be changed to a subheading under 5. Conclusions, instead of presenting it as a standalone section. The same applies to Heading No. 7. Limitations.
The following is just a suggestion; see whether you want to adopt it or not.
- Conclusions – here you provide your traditional conclusion, like what you have done
5.1. Significance and implications of the study – Some of this information is presented on page one before the abstract. See whether you can also bring it here and elaborate further on the significance and implications of the study to both practice and policy.
5.2. Limitations – Information presented here is sufficient. See whether it is also worth reflecting on the fact that since this study was quantitative in nature, it failed to capture the actual verbatim statements from the participants, which could have provided more perspectives. Generally, reflect on the advantages of qualitative research and the shortcomings of quantitative research.
5.3. Recommendations – This information is already presented in future lines of research; see whether you can bring it to this section. Moreover, ensure that this section also responds to all issues indicated in the limitations section, e.g. in response to the limitation of failing to capture the quotations from participants, you can recommend that future studies be conducted using mixed-methods research.
Author Response
Many thanks to the reviewer for their time and dedication. The structure of their work has made our job easier.
We list the questions we have answered.
1.- Statistics provided in lines 53-54 should be referenced.
The location of citations [4,5] has been modified to indicate the source of the cited data.
2.- In line 146, you state, “As the response rate using this method was limited.” It might be worthwhile to clearly indicate the percentage of the response rate and what literature/ theory says is the acceptable response rate.
This refers to the limitation of participation. We expected to achieve 100% of the sample directly from parent organisations, but unfortunately, voluntary participation did not exceed 7% of the questionnaires distributed. We do not know if there are any studies on the level of voluntary participation of parents in this type of study. As our objective was to reach the sample size within a certain time frame, we decided to introduce a second method of recruiting volunteers through school counsellors.
For clarification, we have included this information.
(less than 7% of the questionnaires distributed)
3.- Whilst you state that the study received ethical clearance, you should also discuss the ethical considerations of this study. This information is very important, especially considering the sensitivity of the study.
We have added the following paragraph:
The anonymity of the participating parents was respected at all times. The completed questionnaires and signed informed consent forms were always submitted in sealed envelopes. In this way, we consider that we have respected the confidentiality of the information received. The research team anonymised the computer records and proceeded to destroy the paper documents.
4.- Last comment
Thank you for your suggestions. The structure of the Journal requires that information be placed at the top of the article in order to attract attention and generate interest in the content. For this reason, we cannot modify the general structure
Reviewer 4 Report (New Reviewer)
Comments and Suggestions for Authors
Thank you for your research! My comments to improve your article are:
- The highlights, the main findings and the implications could be moved to Conclusion to better underline the research potential, and to increase structure dynamics.
- A clear literature review section is missing. It seems that the literature review part has been incorporated into Introduction, attenuating the research potential. I suggest that similar studies should be included in an independent literature review section, whereas the reason for conducting a validity and the significance of its venture should be underlined in the Introduction.
- Research assumptions are missing. It is not clear what is the main objective of the validity study. Research assumptions could answer this purpose. In parallel, results could be discussed in light of research assumptions. It is not obvious how results contribute to parental stress management.
- The differences between the mean scores of the American and Spanish samples should be included in Discussion. Conclusions should focus on the research potential and the research contribution to parental stress management, highlighting the behefits of the validity study.
In a nuthsell, your manuscript needs restructure to highlight your important research potential.
Author Response
We thank the reviewer for their comments and the time spent reviewing the article.
1.- The highlights, the main findings and the implications could be moved to Conclusion to better underline the research potential, and to increase structure dynamics.
The journal's editorial team asks us to include the highlights, main conclusions and implications at the beginning of the article. We understand that this is a way of enticing interested readers to decide whether or not to read the article.
2.- A clear literature review section is missing. It seems that the literature review part has been incorporated into Introduction, attenuating the research potential. I suggest that similar studies should be included in an independent literature review section, whereas the reason for conducting a validity and the significance of its venture should be underlined in the Introduction.
In an empirical study, a section on ‘background’ is usually developed as an introduction. The research team has already published a specific systematic review article on the psychometric properties of the PSI-4 [20] Ríos, M., Zerki, S., Alonso-Esteban, Y., & Navarro-Pardo, E. (2022).
3.- Research assumptions are missing. It is not clear what is the main objective of the validity study. Research assumptions could answer this purpose. In parallel, results could be discussed in light of research assumptions. It is not obvious how results contribute to parental stress management.
This is a study on the validity of the translation and adaptation of a tool that assesses parental stress. Without a valid measure, intervention programmes would be aimless. The PSI-4 is a valid instrument for detecting parenting dysfunctions and helps to formulate hypotheses about them. In turn, the PSI-4 can monitor the effects of the intervention.
In order to clarify the use of this tool and, consequently, the objectives of the work, the following sentence has been added at the end of the introduction section:
“The PSI-4 SF is a valid instrument for detecting parenting dysfunctions, while the PSI-4 LF can help to formulate hypotheses about parenting and, thereby, design an intervention programmed. In addition, the PSI-4 can monitor the effects of the intervention”.
4.- The differences between the mean scores of the American and Spanish samples should be included in Discussion. Conclusions should focus on the research potential and the research contribution to parental stress management, highlighting the behefits of the validity study.
The differences between the mean scores of the American and Spanish samples alone justify the need to adapt the scales. Cultural differences mean that parenting behaviours are valued differently on both sides of the Atlantic. A paragraph indicating this point has been added to the discussion section:
“The differences in mean scores between the Spanish and American samples alone justify the need for adaptation, demonstrating how culture can influence the evaluation of parenting behaviours and customs.”
Thanks for your work.
This manuscript is a resubmission of an earlier submission. The following is a list of the peer review reports and author responses from that submission.
Round 1
Reviewer 1 Report
Comments and Suggestions for Authors
I would like to thank the editor for the opportunity to review this article. Below, I provide my detailed suggestions for the authors. Although I appreciate the effort invested in this work, due to concerns regarding novelty and certain methodological nuances that raise doubts about its rigour, I am unable to recommend the manuscript for publication. Nevertheless, I offer constructive recommendations and explain the reasons behind my decision.
The article does not adhere to the latest accepted nomenclature within the field of psychometrics. In addition, the organisation is dense and difficult to follow, particularly the methods section, which is poorly written. Moreover, methodological information appears in the results section, where it should not be included. In this regard, with respect to the types of validity evidence outlined in the most recent edition (2014) of the Standards for Educational and Psychological Measurement—a joint publication of the American Educational Research Association, the American Psychological Association, and the National Council on Measurement in Education, regarded as an authoritative source in social science measurement—modern validity theory no longer aligns with the traditional tripartite view. Messick (1989) argued that ‘construct validity’ encompasses almost all forms of validity evidence, and the unified concept of validity is now widely accepted. The Standards (2014) describe validity as the degree to which all accumulated evidence supports the intended interpretation of test scores for the proposed use, emphasising types of validity evidence rather than discrete types of validity. These five types of validity evidence include: evidence based on test content, response processes, internal structure, relations to other variables, and consequences of testing (Eignor, 2013; Linn, 2011; APA, 2014).
Regarding data analysis, I have identified several methodological concerns that require attention, particularly with respect to statistical analyses and compliance with the COSMIN framework as well as APA and NCME guidelines. These observations are outlined below, with references for consideration:
-
Item distribution analysis is absent; the manuscript does not report floor and ceiling effects nor skewness and kurtosis values for items. These are essential for evaluating item performance and assessing whether the sample distribution approximates normality. Current psychometric standards recommend assessing item distributions first, with acceptable skewness and kurtosis ranging from -1 to 1 (Ferrando & Anguiano-Carrasco, 2010; Muthén & Kaplan, 1985, 1992) or a broader range of -2 to 2 (Bandalos & Finney, 2010; Forero et al., 2009). Omitting these analyses may compromise subsequent validity. Additionally, calculation of item-point biserial correlations is advised to assess item performance.
-
The use of Principal Component Analysis with orthogonal Varimax rotation for exploratory factor analysis is problematic. PCA is not recommended for factor estimation as it ignores measurement error, inflates factor loadings, and overestimates dimensionality (Ferrando & Anguiano-Carrasco, 2010; Gorsuch, 1997a; Vigil-Colet et al., 2009). More appropriate are true factor analysis methods such as EFA with Unweighted Least Squares (ULS), which are computationally efficient.
-
It is unclear whether models with negative variances have been excluded. The presence of items with Heywood cases, where item loadings or higher-order dimensions exceed 1, suggests that analyses are invalid and models may be unstable (Brown, 2006; Ferrando & Anguiano-Carrasco, 2010). Use of Exploratory Structural Equation Modelling (ESEM), a hybrid of EFA and CFA, could be considered.
-
For factor extraction, Horn’s Parallel Analysis is recommended over Kaiser’s rule or scree plots, as it more accurately identifies factors whose eigenvalues exceed chance (Horn, 1965; Lorenzo-Seva et al., 2011).
-
The use of orthogonal Varimax rotation is discouraged in favour of oblique rotations such as Oblimin or Promax, which allow factors to correlate and better represent the interrelated nature of psychological constructs (Fabrigar et al., 1999; Finch, 2006). Studies indicate oblique rotations yield clearer, more interpretable factor structures (Preacher & MacCallum, 2003; Ferrando & Anguiano-Carrasco, 2010).
-
In the Confirmatory Factor Analysis, tau-equivalent models and models with correlated errors should be tested, especially if modification indices exceed 35.000 in the last type of adjustment.
-
Regarding internal consistency, two points are important:
a. Ordinal alpha is preferable to Cronbach’s alpha for Likert-type scales, offering a more precise reliability measure (Zumbo et al., 2007; Gadermann et al., 2012). Its calculation, along with omega coefficients following Doval et al. (2024), would improve assessment.
b. Omega hierarchical coefficients, as proposed by Flora et al. (2020), should be calculated to estimate reliabilities of second- or higher-order factors in hierarchical models. The manuscript appears to have calculated only the standard omega.
Finally, the authors use of an indeterminate correlation for convergent and discriminant validity. If the sample distribution is unknown or non-normal, Spearman’s correlation should be preferred to avoid inflating correlations.
Comments on the Quality of English LanguageThe translation is not rigorous and does not conform to British academic English.
Author Response
Dear reviewer:
We appreciate the effort and time you have devoted to reviewing our work. We accept the methodological limitations described. However, it should be noted that the study presented here reflects the adaptation of an instrument that has already been validated and widely used in clinical practice. Our objective is to demonstrate that the Spanish version of the PSI-4 has at least the same validity as the original version. For this reason, in order to compare the results, we have followed the same methodology as the original study by Abdin (2012).
Regarding specific comments:
- Reviewer's comment:
Item distribution analysis is absent; the manuscript does not report floor and ceiling effects nor skewness and kurtosis values for items. These are essential for evaluating item performance and assessing whether the sample distribution approximates normality. Current psychometric standards recommend assessing item distributions first, with acceptable skewness and kurtosis ranging from -1 to 1 (Ferrando & Anguiano-Carrasco, 2010; Muthén & Kaplan, 1985, 1992) or a broader range of -2 to 2 (Bandalos & Finney, 2010; Forero et al., 2009). Omitting these analyses may compromise subsequent validity. Additionally, calculation of item-point biserial correlations is advised to assess item performance.
Response:
Given the limitations on length of an article, we have chosen to present the results of the distribution analysis in the manual that is currently being written for commercial publication. Furthermore, this is a self-reported test on the feelings of stress that parents experience when raising their children, so the floor and ceiling of the test are determined by a score of 0 and the maximum score obtained by adding the scores of the items, as described in the section on measurement instruments. However, we will take note of this and incorporate the symmetry and kurtosis data for the items in the final publication.
2.- Reviewer's comment:
The use of Principal Component Analysis with orthogonal Varimax rotation for exploratory factor analysis is problematic. PCA is not recommended for factor estimation as it ignores measurement error, inflates factor loadings, and overestimates dimensionality (Ferrando & Anguiano-Carrasco, 2010; Gorsuch, 1997a; Vigil-Colet et al., 2009). More appropriate are true factor analysis methods such as EFA with Unweighted Least Squares (ULS), which are computationally efficient.
Response:
We accept the reviewer's comments. However, we justify the use of principal component analysis with varimax rotation because it was used in the original construction of the instrument (Abidin, 2012) and we only intend to validate the Spanish version. Introducing another methodology could lead to the construction of another model, which would imply another objective.
3.-Reviewer's comment:
It is unclear whether models with negative variances have been excluded. The presence of items with Heywood cases, where item loadings or higher-order dimensions exceed 1, suggests that analyses are invalid and models may be unstable (Brown, 2006; Ferrando & Anguiano-Carrasco, 2010). Use of Exploratory Structural Equation Modelling (ESEM), a hybrid of EFA and CFA, could be considered.
Response:
We agree with the reviewer's opinion and would like to point out that in the section on future lines of research, we consider the possibility of using structural equation models and mediating variables. We believe that the objectives of this article are to adapt an instrument that is used internationally. We believe it is advisable to maintain a Spanish version of the PSI-4 whose measures are comparable to those obtained with the original version. In the future, we will consider explaining the parental stress construct with alternative models.
4.- Reviewer's comment:
For factor extraction, Horn’s Parallel Analysis is recommended over Kaiser’s rule or scree plots, as it more accurately identifies factors whose eigenvalues exceed chance (Horn, 1965; Lorenzo-Seva et al., 2011).
Response
We accept the suggestion for future studies.
5.- Reviewer's comment:
The use of orthogonal Varimax rotation is discouraged in favour of oblique rotations such as Oblimin or Promax, which allow factors to correlate and better represent the interrelated nature of psychological constructs (Fabrigar et al., 1999; Finch, 2006). Studies indicate oblique rotations yield clearer, more interpretable factor structures (Preacher & MacCallum, 2003; Ferrando & Anguiano-Carrasco, 2010).
Response
Indeed, we ourselves point this out as a possibility in the conclusions of the article and future lines of research. The reality is that many of the psychological constructs we evaluate do not respond to orthogonal dimensions. In our case, the use of orthogonal rotation is justified solely by the nature of the study as an adaptation of a previously validated instrument and, consequently, the same original methodology is used.
6.- Reviewer's comment:
In the Confirmatory Factor Analysis, tau-equivalent models and models with correlated errors should be tested, especially if modification indices exceed 35.000 in the last type of adjustment.
Response
We accept the reviewer's suggestion and will take it into account in future studies.
7.- Reviewer's comment:
Regarding internal consistency, two points are important:
a. Ordinal alpha is preferable to Cronbach’s alpha for Likert-type scales, offering a more precise reliability measure (Zumbo et al., 2007; Gadermann et al., 2012). Its calculation, along with omega coefficients following Doval et al. (2024), would improve assessment.
b. Omega hierarchical coefficients, as proposed by Flora et al. (2020), should be calculated to estimate reliabilities of second- or higher-order factors in hierarchical models. The manuscript appears to have calculated only the standard omega.
Response
We agree with the reviewer and have calculated the Alpha and Omega ordinal indices following the methodology indicated in Viladrich, Angulo-Brunet and Doval (2017).
8.- Reviewer's comment:
Finally, the authors use of an indeterminate correlation for convergent and discriminant validity. If the sample distribution is unknown or non-normal, Spearman’s correlation should be preferred to avoid inflating correlations.
Response
The distributions of the scales are normal, so Pearson's correlation was used to calculate convergent validity.
Thank you for your opinions and suggestions, and above all for taking the time to read the article.
Kind regards
Reviewer 2 Report
Comments and Suggestions for Authors
This paper systematically validated the reliability and validity of the Spanish version of the PSI-4, filling the gap caused by the lack of an official standardized tool for Spanish-speaking populations. The study design was rigorous and adopted a multidimensional validity verification framework, including internal consistency, test–retest stability, convergent validity, and discriminant validity. The results confirmed the high reliability and good clinical discriminative ability of the tool, providing an empirical basis for cross-cultural stress research.
Overall, I believe the paper can reach the standard for publication and requires only minor revisions before being accepted. The following are my suggestions for revision. If some issues cannot be addressed, the authors may also mention them in the study’s limitations.
In the sample, 82.1% were mothers, the proportion of fathers was low, and the education level was relatively high (>50% had more than a high school education), which may not fully represent the diversity of family structures in Spain, limiting the generalizability of the results.
The CFA results showed poor fit of the theoretical model (CFI = 0.764, TLI = 0.759), but the reasons were not explored in depth, nor were alternative models attempted; the analysis relied solely on partial support from the EFA, lacking substantive examination of model modifications.
The Methods section mentions that the PSI-4 includes the Life Stress subscale, but its reliability and validity data were not reported in the Results section.
The convergent validity analysis was based only on a subsample of 87 parents, and the test–retest reliability sample size was too small, which may overestimate the stability of the correlations.
Author Response
Dear reviewer:
Thank you for your time and dedication in reviewing our paper.
Regarding your comments:
- a) The sample may be biased and not adequately represent the diversity of family structures in Spain. Given the characteristics of the instrument (self-report), respondents need to have a good level of reading comprehension, which excludes parents with lower levels of education. However, we take note of this limitation and will indicate this in the instrument's user manual.
- b) Regarding the results of the Confirmatory Factor Analysis, alternative models have not been considered. This is a study adapting an instrument used worldwide. Altering the model would possibly involve creating a new instrument whose results would not be comparable across cultures. For this reason, we retraced the steps used to construct the instrument and used the same methodology as the original in Abdin (2012)
- c) Regarding the inclusion of the complementary Vital Stress Scale. As it is complementary and not part of the parental stress index, it was excluded from the analysis. We have corrected the text of the description to avoid confusion.
- d) Regarding convergent validity and test-retest reliability, we accept the limitations derived from the sample sizes. A final point has been added to the paper highlighting these limitations.
Finally, we would like to thank the reviewer for their comments, which will undoubtedly lead to an improvement in the content of the article.
Yours sincerely
Round 2
Reviewer 1 Report
Comments and Suggestions for Authors
I appreciate the responses provided by the authors; nevertheless, I would like to set out my perspective below.
In their replies, the authors have not offered a robust and scientifically substantiated argument regarding the methodological issues raised about the currency of the methods employed. They merely justify the use of outdated procedures on the grounds of replicating the original study, without taking into account the substantial international evidence demanding the application of more advanced psychometric standards, as cited in my previous review (APA, NMCE, among others). I also add Fenn (2020), who supports my perspective. According to recent literature, “the emphasis on ethical frameworks, transparency, and the need for continuous validation of tests will shape the effectiveness of psychometric assessments in various domains… By adhering to these updated international norms, practitioners can ensure that their assessments not only measure aptitude and personality traits accurately but also uphold the dignity and rights of all individuals involved in the testing process” (Fenn, 2020, p.134). This quotation underscores that adherence to updated international standards is essential for the validity of instruments and the preparation of current scientific articles.
Further, contemporary guidelines specify: “Modern psychometrics employs advanced statistical models to identify and revise biased items, ensuring assessments measure the intended construct rather than extraneous factors” (Flora et al., 2020), thus excluding validation through mere replication of obsolete methodologies. The use of robust factor analysis is considered the minimal standard in current practice, achieving improvements of up to 30% in accuracy and validity compared to traditional methods such as PCA with varimax. “Adhering to best practice and updating methods is critical for reliable, valid, and reproducible psychometric assessment” (Fenn, 2020).
In this regard, I wish to stress that addressing the methodological issues raised merely by mentioning certain analyses in the limitations section or as avenues for future research is neither sufficient nor appropriate, especially when such analyses are wholly feasible given the current state of psychometrics. Moreover, the CFAs conducted continue to exhibit Heywood cases—that is, improper solutions such as negative error variances—which result in invalid and unstable models and demonstrate the necessity of employing more rigorous and up-to-date methodologies. The continued insistence on replicating outdated methodologies, whilst disregarding these advances and explicit recommendations, constitutes a serious omission in relation to the scientific principles of transparency, responsibility, and continual revision. Otherwise, biases are perpetuated and the quality of the resulting evidence is compromised, contrary to international best practice.
Furthermore, the manuscript still does not adopt the terminology or adhere to the guidelines set forth by leading bodies such as the American Psychological Association (APA), COSMIN and the National Council on Measurement in Education (NCME). This falls short of international expectations for quality, transparency, and terminological rigour in contemporary psychometric publications.
Comments on the Quality of English LanguageMust be improved.
Author Response
Dear reviewer:
Thank you for taking the time to review our work. It has been very useful to us. We initially thought that in order to compare the original version with the Spanish version, we should follow the same methodology used to construct the PSI-4. We appreciate your comments, we accept them, and we have modified the analyses. Instead of following the original model, we have opted to develop two Exploratory Factor Analyses (one for each of the two domains of the instrument). The EFA model used was WLSMV (Weighted Least Squares Mean and Variance adjusted). We have made the corresponding changes.
We hope we have answered your questions.
Best regards